# Interferon Regulatory Factor 5 (*IRF5*) Gene Haplotypes Are Associated with Premature Coronary Artery Disease. Association of the *IRF5* Polymorphisms with Cardiometabolic Parameters. The Genetics of Atherosclerotic Disease (GEA) Mexican Study

**DOI:** 10.3390/biom11030443

**Published:** 2021-03-17

**Authors:** Rosalinda Posadas-Sánchez, Guillermo Cardoso-Saldaña, José Manuel Fragoso, Gilberto Vargas-Alarcón

**Affiliations:** 1Department of Endocrinology, Instituto Nacional de Cardiología Ignacio Chávez, 14080 Mexico City, Mexico; rossy_posadas_s@yahoo.it (R.P.-S.); gccardosos@yahoo.com (G.C.-S.); 2Department of Molecular Biology, Instituto Nacional de Cardiología Ignacio Chávez, 14080 Mexico City, Mexico; mfragoso1275@yahoo.com.mx

**Keywords:** cardiometabolic parameters, genetic association, interferon regulatory factor 5, polymorphisms, premature coronary artery disease

## Abstract

Interferon regulatory factor 5 (*IRF5*) has an important role in the inflammatory process, a fundamental component of coronary artery disease (CAD). Thus, the objective of this study was to evaluate the association of *IRF5* polymorphisms with the development of premature CAD (pCAD) and cardiometabolic parameters. *IRF5* polymorphisms (rs1874330, rs3778754, rs3757386, rs3757385, rs3807134, rs3807135, and rs6968563) were determined in 1116 pCAD patients and 1003 controls. Polymorphism distribution was similar in patients and controls; however, the haplotype analysis showed five haplotypes with a different distribution. *TGCGTCT* (OR (odds ratio) = 1.248, *p* = 0005) and *TCTGCCT* (OR = 10.73, *p* < 0.0001) were associated with a high risk, whereas *TCCGTCT* (OR = 0.155, *p* < 0.0001), *CGCTTTT* (OR = 0.108, *p* < 0.0001), and *TCCGCCT* (OR = 0.014, *p* < 0.0001) were associated with a low risk of pCAD. Associations with aspartate aminotransferase, hypertriglyceridemia, magnesium deficiency, triglycerides/HDL-C index, LDL-C, and adiponectin levels were observed in pCAD patients. In controls, associations with hypoalphalipoproteinemia, non-HDL-C, apolipoprotein B, hyperuricemia, TNF-α, IL-6, IL-15, valvular calcification, and subclinical hypothyroidism were observed. In summary, five haplotypes were associated with pCAD, two with high risk and three with low risk. Some *IRF5* polymorphisms were associated with cardiometabolic parameters in pCAD patients and control.

## 1. Introduction

Coronary artery disease (CAD), a clinical manifestation of atherosclerosis, is one of the leading causes of death and morbidity worldwide [1]. The role of inflammation is well known in the progression of atherosclerosis. The infiltration of several types of cells, including T cells and macrophages, in the atherosclerotic plaque has been reported [2]. The production of cytokines, chemokines, and growth factors by these cells perpetuates the damage in the atherosclerotic lesion [3]. Interferon regulatory factor 5 (*IRF5*) plays a central role in inflammation, mediating the production of proinflammatory cytokines, such as IL-6, IL-12, IL-23, and TNF-α [4,5]. *IRF5* is expressed in monocytes and macrophages and has an important role in defining the inflammatory macrophage phenotype [5]. It has been reported that *IRF5* participates in the Akt2 activation, producing an increase of glycolysis and M1 macrophage polarization [6]. M1 macrophages can produce proinflammatory cytokines and are suggested to be involved in the development of atherosclerosis and affect the stabilization and impact of atherosclerotic plaques [7,8]. Seneviratne et al., using an animal model, established that *IRF5* promotes the presence of proinflammatory CD11c+ macrophages within atherosclerotic lesions and controls the expansion of the necrotic core formation in atherosclerosis [9]. In murine models, it has been reported that the inhibition of *IRF5* decreases myocardial infarction size, and its genetic deletion protects from inflammatory arthritis and insulin resistance [10,11,12]. The *IRF5* gene is located in chromosome 7 in location 128,937,457–128,950,038 (ENSEMBL genome browser) and is highly polymorphic. *IRF5* polymorphisms have been associated with the presence of coronary artery calcium in patients with systemic lupus erythematosus [13]. In another study, it was reported that *IRF5* is expressed in cells of atherosclerotic tissue and that this expression is regulated by some *IRF5* polymorphisms; however, these polymorphisms were not associated with CAD or related phenotypes [14]. Considering the important role of *IRF5* in the atherosclerotic process suggested previously, the present study aimed to establish the association of *IRF5* polymorphisms with the presence of pCAD and/or cardiometabolic parameters. Knowing genetic polymorphisms associated with the development of CAD can help select patients with high genetic risk or establish which therapy is the most appropriate for a specific patient. To our knowledge, our study is the first in which an association between *IRF5* polymorphisms and the presence of CAD is established. After performing informatics analysis, seven polymorphisms with a minor allele frequency >5% and/or with probable functional effect were included in the study.

## 2. Materials and Methods

### 2.1. Ethics Statement

The study was approved by the Ethics Committee of the Instituto Nacional de Cardiología Ignacio Chávez (INCICH). All participants gave written informed consent, and the protocol followed the recommendations of the Declaration of Helsinki.

### 2.2. Subjects

The study included 1116 patients with pCAD and 1003 healthy controls, all of them belonging to the Genetics of Atherosclerotic Disease (GEA) Mexican Study. Premature CAD was defined as a history of myocardial infarction, revascularization surgery, angioplasty, and coronary stenosis >50% on angiography. Premature CAD was considered when the diagnosis was made before age 55 in men and before age 65 in women. The control group was recruited from the institute’s blood bank and by direct invitation. This group included healthy individuals with no family history of pCAD. Patients and controls were recruited regardless of the comorbidities they presented. Thus, comorbidities were not considered as an exclusion criterion in the study. In all individuals, a computed tomography (CT) of the chest and abdomen was performed. Total, subcutaneous, and visceral abdominal fat areas were quantified as described by Kvist et al. [15], and the coronary artery calcification (CAC) score using the Agatston method [16]. All individuals included in the control group presented a CAC score equal to zero. Demographic, clinical, and biochemical parameters and lifestyle characteristics were evaluated in all participants and defined as previously described [17,18,19,20,21,22].

To assess the possible influence of population stratification, a panel of 265 ancestry informative markers distinguishing mainly Amerindian, European, and African ancestries were determined in all individuals [23]. A similar global ancestry was observed in the study individuals with 54.0% of Amerindian, 35.8% of Caucasian, and 10.1% of African ancestry in controls and 55.8% of Amerindian, 34.3% of Caucasian, and 9.8% of African ancestry in pCAD patients [18].

### 2.3. Genetic Analysis

High-molecular-weight genomic DNA was extracted from peripheral blood using the QIAamp DNA Blood Mini kit (QIAGEN, Hilden, Germany). The possible functional effect of the *IRF5* SNPs was evaluated using the SNP Function Prediction (http://snpinfo.niehs.nih.gov/snpinfo/snpfunc.html, accessed on 15 March 2020), Splice Port: An Interactive Splice Site Analysis Tool (http://spliceport.cbcb.umd.edu/SplicingAnalyser.html, accessed on 15 March 2020), Human-Transcriptome Database for Alternative Splicing (http://www.h-invitational.jp/h-dbas/, accessed on 15 March 2020), HSF (http://www.umd.be/HSF/, accessed on 15 March 2020), ESE finder (http://rulai.cshl.edu/cgi-bin/tools/ESE3/esefinder.cgi, accessed on 15 March 2020), and SNPs3D (http://www.snps3d.org/, accessed on 15 March 2020) bioinformatics tools. After this functional analysis, we selected for the study seven *IRF5* polymorphisms (rs1874330, rs3778754, rs3757386, rs3757385, rs3807134, rs3807135, and rs6968563) that were determined using 5′ exonuclease TaqMan genotyping assays. The polymorphisms were genotyped on an ABI Prism 7900HT Fast Real-Time PCR System (Applied Biosystems, Foster City, CA, USA). To corroborate the adequate assignment of the genotypes in the TaqMan assays, 10% of the samples were randomly selected and repeated. These samples were 100% concordant in two independent assays.

### 2.4. Statistical Analysis

Data are expressed as frequencies, median (interquartile range), or mean ± standard deviation, as appropriate. Either Mann–Whitney U or Student’s *t*-test was used for continuous variable comparisons, while the chi-square test was employed for categorical variable comparisons. Alleles and genotype frequencies were determined by direct counting. Hardy–Weinberg’s equilibrium was determined by the chi-square test. The association of the polymorphisms with pCAD and with cardiometabolic parameters was evaluated using logistic regression analysis under different inheritance models (additive, codominant 1, codominant 2, dominant, heterozygote, and recessive). The different models were adjusted for confounding variables as appropriate. Haploview version 4.1 (https://www.broadinstitute.org/haploview/haploview, accessed on 15 October 2020) (Broad Institute of Massachusetts Institute of Technology and Harvard University, Cambridge, MA, USA) was used to establish linkage disequilibrium (LD, D’) and construction of haplotypes.

## 3. Results

### 3.1. Demographic, Clinical, Biochemical, and Lifestyle Characteristics

The analysis included 2119 individuals, 1116 with pCAD and 1003 healthy controls (CAC score equal to zero). Demographic, clinical, biochemical, and lifestyle characteristics in the studied groups are shown in Table 1.

Compared with controls, body mass index (BMI), triglycerides, triglycerides/HDL-C index, aspartate aminotransferase (AST), uric acid, and interleukin 6 were higher in patients with pCAD. In the same way, the pCAD patients showed a high frequency of obesity, hypertension, type 2 diabetes mellitus, hypoalphalipoproteinemia, hypertriglyceridemia, triglycerides/HDL-C index, hypoadiponectinemia, hyperuricemia, and magnesium deficiency when compared with the healthy controls. In contrast, the patients showed lower levels of LDL cholesterol, apolipoprotein B, and non-HDL cholesterol. This decrease may be due to the statin treatment that the patients received (Table 2).

High non-high-density lipoprotein cholesterol (non-HDL-C) was defined when its values were >160 mg/dL. Increased TNF-α, IL-6, and IL-15 were defined as follow: TNF-α > 75th percentile (0.97 pg/mL in women and 2.13 pg/mL in men); IL-6 > 75th percentile (1.30 pg/mL in women and 1.78 pg/mL in men); IL-15 > 75th percentile (2.02 pg/mL in women and 2.41 pg/mL in men). These cutoff points were obtained from a GEA Mexican study sample of 131 men and 185 women without obesity and with normal values of blood pressure, fasting glucose, and lipids.

### 3.2. Association of IRF5 Polymorphisms with pCAD

The seven polymorphisms evaluated independently were not associated with the risk of premature pCAD (data not shown).

### 3.3. Distribution of IRF5 Haplotypes in pCAD Patients and Healthy Controls

After the linkage disequilibrium analysis, eight haplotypes were formed. Table 3 shows the distribution of haplotypes in pCAD patients and healthy controls. *TGCGTCT* (OR (odds ratio) = 1.248, 95% CI: 1.102–1.413, *p* = 0005) and *TCTGCCT* (OR = 10.73, 95% CI: 5.416–21.26, *p* < 0.0001) were associated with a high risk, whereas *TCCGTCT* (OR = 0.155, 95% CI: 0.089–0.269, *p* < 0.0001), *CGCTTTT* (OR = 0.108, 95% CI: 0.051–0.226, *p* < 0.0001), and *TCCGCCT* (OR = 0.014, 95% CI: 0.002–0.102, *p* < 0.0001) were associated with a low risk of pCAD when compared with healthy controls.

### 3.4. Association of IRF5 Polymorphisms with Cardiometabolic Parameters

The associations of *IRF5* polymorphisms with cardiometabolic parameters were evaluated independently in pCAD patients and healthy controls. In healthy controls, rs3757385 (OR = 0.63, 95% CI: 0.46–0.85, *p*_dominant_ = 0.002) and rs3807135 (OR = 0.67, 95% CI: 0.49–0.92, *p*_codominant 1_ = 0.013) were associated with a low risk of hypoalphalipoproteinemia, rs3807134 with elevated non-HDL-cholesterol (OR = 1.78, 95% CI: 1.12–2.82, *p*_heterozygote_ = 0.015), rs6968563 with elevated apolipoprotein B (OR = 2.04, 95% CI: 1.07–3.88, *p*_heterozygote_ = 0.030), rs3807134 with a high risk of hyperuricemia (OR = 2.07, 95% CI: 1.24–3.43, *p*_codominant 1_ = 0.005), and rs3778754 with elevated levels of TNF-α (OR = 1.51, 95% CI: 1.08–2.11, *p*_recessive_ = 0.017). In the same way, rs1874330 (OR = 0.72, 95% CI: 0.53–0.98, *p*_recessive_ = 0.038) and rs3807135 (OR = 0.69, 95% CI: 0.49–0.96, *p*_recessive_ = 0.030) were associated with a low risk to have elevated IL-6 levels, rs1874330 (OR = 0.71, 95% CI: 0.53–0.96, *p*_recessive_ = 0.023) with a low risk to have elevated IL-15 levels, rs3778754 (OR = 1.565, 95% CI: 1.12–2.16, *p*_additive_ = 0.009) with a high risk of valvular calcification, and rs3757385 and rs3807135 with a high risk of subclinical hypothyroidism (OR = 1.73, 95% CI: 1.20–2.51, *p*_recessive_ = 0.004) (Figure 1). The models were adjusted by age, sex, and body mass index.

In the patient group, four polymorphisms (rs1874330, rs3778754, rs3757385, and rs3807135) were associated with low AST levels, high LDL-C levels, and a low risk of magnesium deficiency. Three of them (rs3778754, rs3757385, and rs3807135) were associated with a low risk of hypertriglyceridemia, and four (rs3757386, rs3757385, rs3807134, and rs3807135) were associated with elevated triglycerides/HDL-cholesterol index. Finally, two polymorphisms (rs1874330 and rs3757385) were associated with a high risk of having low adiponectin levels (Figure 2). The models were adjusted by age, sex, and body mass index. A summary of results is shown in Figure 3.

## 4. Discussion

Atherosclerosis is a chronic and progressive disease that begins early in life and is characterized by a long subclinical phase, which progresses, producing coronary artery disease. Unfortunately, atherosclerosis is frequently diagnosed in advanced stages and generally after a sudden and sometimes fatal cardiovascular event. Genetic background is involved in both the triggering and progression of atherosclerosis [24]. Thus, we analyzed the distribution of seven polymorphisms of the *IRF5* gene in patients with pCAD and healthy controls. The distribution of the polymorphisms was similar in the study groups; however, different haplotypes were associated with pCAD. *TGCGTCT* and *TCTGCCT* were associated with a high risk, and *TCCGTCT*, *CGCTTTT*, and *TCCGCCT* with a low risk of pCAD. In the same way, some polymorphisms were associated with cardiometabolic parameters.

*IRF5* is an important regulator of the production of pro-inflammatory cytokines, such as TNF-alpha, IL-6, IL-12, and IL-23 [4,5]. Given these effects, *IRF5* is an important regulator of inflammation. In the same way, *IRF5* has been reported to modulate the genotype and function of macrophages, affecting the formation and stability of atherosclerotic plaque [9]. Mälarstig et al. reported that *IRF5* is expressed in cells in atherosclerotic plaques and that some polymorphisms located in the *IRF5* gene modified this expression [14]. In this study, the authors determined 10 polymorphisms, none of which were associated with CAD. Of those polymorphisms, in our study only one of them was included (rs3757385). The selection of the polymorphisms for the Mälarstig study was made considering those polymorphisms previously associated with some autoimmune diseases and with expression levels. In our case, the polymorphisms were selected using informatics tools. Similar to the Mälarstig study, we did not detect associations of the polymorphisms studied with pCAD. However, when the haplotype analysis was made, some haplotypes were associated with pCAD. It has been suggested that the use of haplotypes could be a better tool to capture more relevant information in a specific region compared with the analysis of independent polymorphisms [25]. The combination of polymorphisms in a haplotype could have a greater effect on the genotype of interest, and this effect would be greater than when analyzing the polymorphisms independently [26].

Association of the polymorphisms with cardiometabolic parameters was made independently in pCAD patients and healthy controls. In patients, the association of the polymorphisms with AST, LDL-C, magnesium deficiency, hypertriglyceridemia, triglycerides/HDL-C index, and adiponectin levels is worth mentioning. It is important to note that the rs3757385 polymorphism was associated with six cardiometabolic variables in this group of patients. In the study of Mälarstig et al., this polymorphism was not associated with CAD; however, it was significantly associated with *IRF5* mRNA expression levels in carotid plaques [14]. This polymorphism is located in the promoter region of the gene and, according to the informatics analysis, is a tag SNP. The change in this position produces binding sites for BCL6 (G allele), STAT (G allele), and YY1 (T allele) transcription factors. This polymorphism has been associated with interstitial lung disease associated with systemic sclerosis [27,28] and unexplained recurrent pregnancy loss [29]. On the other hand, in controls, the polymorphisms were associated with hypoalphalipoproteinemia, non-HDL cholesterol, apolipoprotein B, hyperuricemia, TNF-α, IL-6, IL-15, valvular calcification, and subclinical hypothyroidism. As can be seen, the associations detected were different in both study groups, which is to be expected because these two groups are different. It is noteworthy that the control group does not have a family history of cardiovascular disease. Sindhu et al. reported that adipose tissue *IRF5* gene expression was associated with cardiometabolic parameters in diabetic obese patients. A positive correlation was reported for LDL-C, HDL-C, triglycerides, TNF-α, and IL-6 [30], variables associated with *IRF5* polymorphisms in our study. In an animal model of *IRF5*-deficient systemic lupus erythematosus, an increase in atherosclerosis was observed. This was accompanied by metabolic disturbances, such as insulin resistance, hyperglycemia, hyperlipidemia, increased adiposity, and hepatic steatosis [31].

The detection of polymorphisms associated with the development of CAD using either genome-wide association studies or candidate gene studies can be very important for the detection of individuals at high risk of developing CAD even before they present symptoms. Early detection can help define better and more targeted treatments. In the same way, in the context of precision medicine, knowledge of these polymorphisms can contribute to better individual therapy with beneficial results for affected patients.

Our study has several strengths, among which the inclusion of a large number of patients and controls well characterized from the demographic, clinical, and biochemical points of view stands out. This allowed us to analyze the association of polymorphisms with pCAD and with cardiometabolic variables. In the same way, it allowed us to adjust the inheritance models for confounding factors that are directly associated with the disease.

Among the limitations, we can mention that the possible functional effect was only evaluated with bioinformatics tools, and an experimental design was not included. Additionally, in our study, it was not possible to measure the levels of expression of *IRF5*. Finally, the results were not replicated in an independent group of individuals with and without pCAD.

## 5. Conclusions

In summary, five haplotypes were associated with pCAD, two with a high risk (*TGCGTCT* and *TCTGCCT*) and three with a low risk (*TCCGTCT*, *CGCTTTT*, and *TCCGCCT*). Some *IRF5* polymorphisms were associated with cardiometabolic parameters in pCAD patients and healthy controls (Figure 3).

## Figures and Tables

**Figure 1 biomolecules-11-00443-f001:**
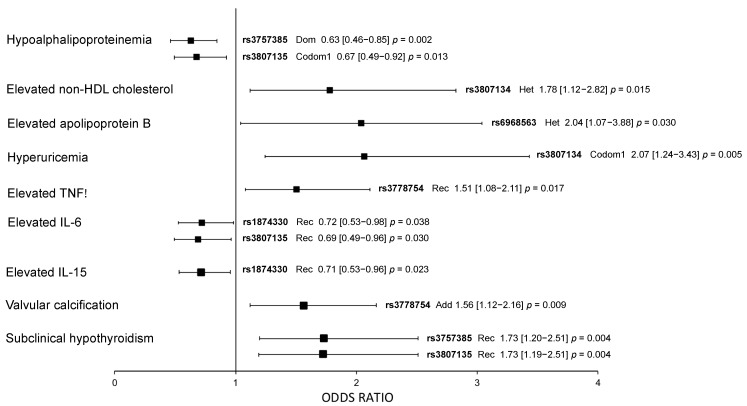
Association of *IRF5* polymorphisms with cardiometabolic parameters in healthy controls. All models were adjusted by age, sex, and BMI. Dom: dominant model; Rec: recessive model; Codom1: codominant 1 model; Het: heterozygote model; Add: additive model. OR (95% CI) *p* values.

**Figure 2 biomolecules-11-00443-f002:**
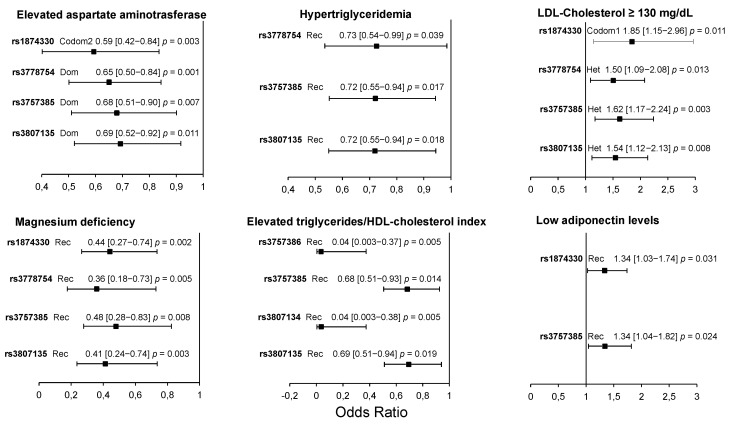
Association of *IRF5* polymorphisms with cardiometabolic parameters in pCAD patients. All models were adjusted by age, sex, and BMI. Dom: dominant model; Rec: recessive model; Codom1: codominant 1 model; Codom2: codominant 2 model; Het: heterozygote model. OR (95% CI) *p* values.

**Figure 3 biomolecules-11-00443-f003:**
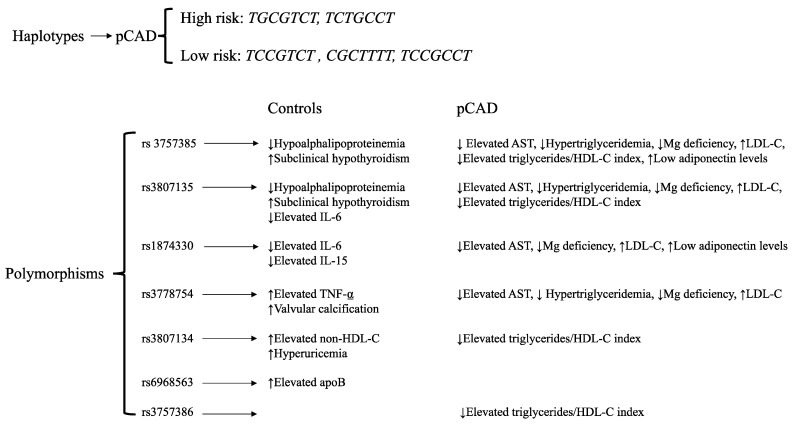
Summary results. AST: aspartate aminotransferase.

**Table 1 biomolecules-11-00443-t001:** Demographic, clinical, biochemical, and lifestyle characteristics in the study groups.

	Study Groups		
	Controls (*n* = 1003)	pCAD (*n* = 1116)	*p* *
*Demographic and clinical characteristics*			
Age (years)	51 ± 9	54 ± 8	<0.001
Sex (% male)	41.6	81.5	<0.001
Body mass index (kg/m^2^)	27.9 (25.4–30.7)	28.3 (26.0–31.1)	0.004
*Biochemical profile*			
HDL-cholesterol (mg/dL)	45 (36–65)	37 (31–44)	<0.001
LDL-cholesterol (mg/dL)	115 (95–134)	91 (68–116)	<0.001
Triglycerides (mg/dL)	144 (107–203)	162 (120–219)	<0.001
Non-HDL-cholesterol (mg/dL)	142 (121–164)	120 (93–151)	<0.001
Apolipoprotein B	94 (76–113)	80 (63–103)	<0.001
Triglycerides/HDL-cholesterol	3.2 (2.1–5.3)	4.3 (3.0–6.6)	<0.001
Aspartate aminotransferase (UI)	24 (21–30)	26 (22–31)	0.001
Adiponectin	8.2 (5.0–12.6)	5.2 (3.2–8.1)	<0.001
Uric acid (mg/dL)	5.4 (4.4–6.4)	6.4 (5.4–7.4)	<0.001
Tumor necrosis factor alpha (pg/mL)	0.56 (0.01–1.81)	0.53 (0.06–1.65)	0.377
Interleukin 6 (pg/mL)	0.83 (0.40–1.71)	0.93 (0.50–2.04)	0.011
Interleukin 15 (pg/mL)	1.46 (0.34–2.94)	1.30 (0.67–2.06)	0.022
*Lifestyle*			
Current smoking habit (%)	23.3	11.6	<0.001
Physical activity	7.9 (7.0–8.8)	7.5 (6.8–8.4)	<0.001

Data are shown as mean ± standard deviation, median (interquartile range), or percentage. * Student’s *t*-test, Mann–Whitney’s U test, or chi square test.

**Table 2 biomolecules-11-00443-t002:** Prevalence of coronary risk factors in the study groups.

	Study Groups		
	Controls (*n* = 1003)	pCAD (*n* = 1116)	*p* *
Obesity (%)	29.7	34.9	0.012
Hypertension (%)	18.7	68.0	<0.001
Type 2 diabetes mellitus (%)	10.3	35.5	<0.001
Hypoalphalipoproteinemia	51.4	67.8	<0.001
High LDL-cholesterol (≥130 mg/dL, %)	29.3	16.5	<0.001
Hypertriglyceridemia (%)	33.2	42.8	<0.001
High non-HDL-cholesterol (>160 mg/dL, %)	27.9	19.6	<0.001
High apolipoprotein B (≥110 mg/dL, %)	28.2	19.5	<0.001
High triglycerides/HDL-cholesterol index (>3.0, %)	52.6	75.2	<0.001
Elevated aspartate aminotransferase (%)	35.9	38.3	0.259
Hypoadiponectinemia (<p25, %)	42.7	58.1	<0.001
Hyperuricemia (%)	20.4	36.0	<0.001
Elevated TNFα (>p75, %)	29.9	23.3	0.001
Elevated interleukin 6 (>p75, %)	29.2	29.8	0.803
Elevated interleukin 15 (>p75, %)	36.5	19.4	<0.001
Magnesium deficiency (%)	5.0	9.3	<0.001
Valvular calcification (%)	10.6	nd	
Subclinical hypothyroidism (%)	17.3	16.3	0.592

Data are shown as percentages. * Chi square test. nd: not determined.

**Table 3 biomolecules-11-00443-t003:** *IRF5* haplotype frequencies and the presence of pCAD.

Haplotypes		pCAD	Controls	OR (95% CI)	*p*
H1	*TGCGTCT*	0.427	0.374	1.248 (1.102–1.413)	0.0005
H2	*CCCTTTT*	0.407	0.379	1.128 (0.996–1.278)	0.0569
H3	*TCCTTTT*	0.061	0.048	1.285 (0.980–1.684)	0.0700
H4	*TCTGCCT*	0.047	0.004	10.73 (5.416–21.26)	<0.0001
H5	*TCCGTCT*	0.007	0.042	0.155 (0.089–0.269)	<0.0001
H6	*TCCGTCC*	0.023	0.018	1.303 (0.843–2.012)	0.2331
H7	*CGCTTTT*	0.004	0.032	0.108 (0.051–0.226)	<0.0001
H8	*TCCGCCT*	0.001	0.031	0.014 (0.002–0.102)	<0.0001

OR, odds ratio; CI, confidence interval. The order of the polymorphisms in the haplotype is according to the position in the chromosome (rs1874330, rs3778754, rs3757386, rs3757385, rs3807134, rs3807135, and rs6968563).

## Data Availability

**Informed**: The data presented in this study are available upon request from the corresponding author.

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
