# Peer review of "Interferon Regulatory Factor 5 (IRF5) Gene Haplotypes Are Associated with Premature Coronary Artery Disease. Association of the IRF5 Polymorphisms with Cardiometabolic Parameters. The Genetics of Atherosclerotic Disease (GEA) Mexican Study"

_biomolecules, 2021, doi:10.3390/biom11030443_

Round 1
Reviewer 1 Report
Please highlight the novelty of the current study in the introduction
Does the comorbidities of the participants recruited be considered for study inclusion and exclusion? Do patients take medications? Will this interfere with the biochemical profile?
Can the authors give any suggestion about the potential clinical translation of their data? Should we perform genetic screening for early diagnosis and prevention?
line 82-84, please rephrase the sentence
line 33 Coronary artery disease - change C to c
line 126 chi square test - 't' is missing
Author Response
1.- Please highlight the novelty of the current study in the introduction
Answer: Our study is the first that reported a positive association of IRF5 polymorphisms with the presence of CAD. This novelty is commented on in the introduction section. The phrase “Knowing genetic polymorphisms associated with the development of CAD can help select patients with high genetic risk or establish which therapy is the most appropriate for a specific patient. To our knowledge, our study is the first in which an association between IRF5polymorphisms and the presence of CAD is established.” has been included in the introduction section.
2.- Does the comorbidities of the participants recruited be considered for study inclusion and exclusion? Do patients take medications? Will this interfere with the biochemical profile?
Answer: Comorbidities were not considered as an exclusion criterion in the study. The phrase “Patients and controls were recruited regardless of the comorbidities they presented. Thus, comorbidities were not considered as an exclusion criterion in the study.” has been added in the material and methods section.
Patients were taking medications and these can alter biochemical profiles. In order to clarify this point, the phrase “In contrast, the patients showed lower levels of LDL cholesterol, apolipoprotein B, and non-HDL cholesterol. This decrease may be due to the statin treatment that patients received.” has been added in the results section.
3.- Can the authors give any suggestions about the potential clinical translation of their data? Should we perform genetic screening for early diagnosis and prevention?
Answer: In order to clarify this point, the phrase “The detection of polymorphisms associated with the development of CAD either using genome-wide association studies or candidate gene studies can be very important for the detection of individuals at high risk of developing CAD even before they present symptoms. Early detection can help define better and more targeted treatments. In the same way, in the context of precision medicine, knowledge of these polymorphisms can contribute to better individual therapy having beneficial results for affected patients.” has been added in the discussion section.
4.- line 82-84, please rephrase the sentence
Answer: The phrase has been corrected
“To assess the possible influence of population stratification, a panel of 265 ancestry informative markers distinguishing mainly Amerindian, European, and African ancestry were determined in all individuals”
5.- line 33 Coronary artery disease - change C to c
Answer: It has been corrected
6.- line 126 chi square test - 't' is missing
Answer: It has been corrected.
Reviewer 2 Report
I have no major objections to this work, I only think that it would be suitable if authors would create and include summary/central figure showing the most robust polymorhphisms/five haplotypes of interferon regulatory factor 5 that are linked to premature CAD. It would be beneficial to convey message this way to the general audience.
Author Response
Answer: As is suggested by the reviewer, a figure (Figure 3) has been included that summarizes the results of the study.